# FOUNDATION POLICIES WITH MEMORY

## ABSTRACT

A generalist agent should perform well on novel tasks in unfamiliar environments. While Foundation Policies (FPs) enable generalization across new tasks, they lack mechanisms for handling novel dynamics. Conversely, agents equipped with memory models can adapt to new dynamics, but struggle with unseen tasks. In this work, we bridge this gap by integrating memory models into the FP architecture, allowing policies to condition on both task and environment dynamics. We evaluate FPs enhanced with attention, state-space, and RNN-based memory models on POPGym, a memory benchmark, and ExORL, an unsupervised RL benchmark. Our results show that GRUs achieve the best generalization to unseen tasks and dynamics for a given recurrent state size, approaching the performance of a supervised baseline that has access to task information during training and significantly outperforming memory-free FPs. Additionally, our approach improves FP performance on entirely new environments not encountered during training. Our anonymized code is available at `https://anonymous.4open.science/r/zero-shot-96A1`, and our datasets are open-sourced at REDACTED.

## 1 INTRODUCTION

Reinforcement Learning (RL) agents [92] exhibit superhuman decision-making skill when tasked with a *single* objective in a *single* environment [89, 67, 90, 91]. A new line of work focuses on producing *generalist agents* that replicate such results across *many* tasks and environments [83, 53, 105]. Foundation policies (FPs) [96, 97, 76, 45] are a promising approach for building generalist agents, providing a principled mechanism for generalising to *any* downstream task in an environment after an offline reward-free pre-training phase. However, as yet, FPs are not equipped to deal with a change in dynamics between pre-training and deployment.

A concurrent line of work on *in-context* RL attempts to build generalist agents by using *memory models* to condition policies on reward-labelled trajectories [14, 43, 56, 54, 62, 23] or to reach arbitrary goal states [31]. In principal, these models can perform dynamics generalisation by inferring changes between training and testing from the trajectory used to condition the policy. However, they lack the task generalisation ability of FPs for two reasons. They are either 1) trained with reward supervision and so cannot reliably generalise to new tasks with different reward functions, or are 2) trained without reward supervision to reach any goal-state in an environment and so cannot reliably generalise to new tasks not codified by a goal state.

Here, we reconcile these lines of work and propose *foundation policies with memory*, an architecture that, like in-context RL agents, infers the current dynamics context using powerful memory models and passes it to an FP for solving unseen tasks. We evaluate FPs with attention [98], state-space [32, 33], and RNN-based [24, 17] memory models across a range of experiments testing their ability to infer the dynamics context, and generalise to unseen tasks in unseen dynamics. We find that GRUs achieve the best generalisation to unseen tasks and dynamics for a given recurrent state size, approaching the performance of a supervised baseline that has access to task information during training and significantly outperforming memory-free FPs (Figure 1). Finally, we find that FPs with memory improve FP performance on entirely new environments not seen during training.

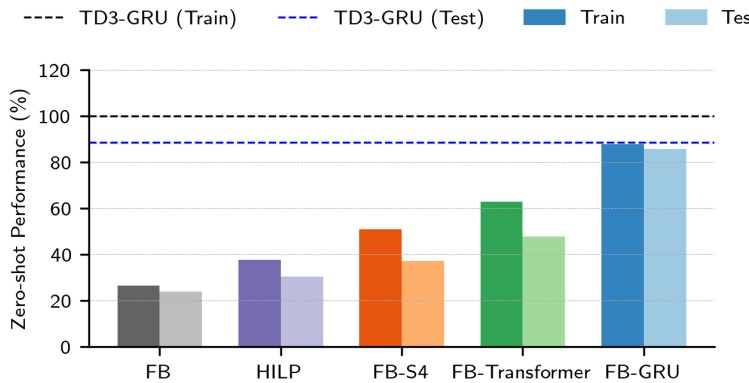

Figure 1: **Zero-shot task and dynamics generalisation.** FPs with memory models generalise to test tasks and dynamics not seen during training on the ExORL benchmark. FB-GRU approaches the performance of a supervised baseline, TD3-GRU, despite being trained without rewards. A full discussion is provided in Section 4.3.

## 2 PRELIMINARIES

**Contextual markov decision processes.** A Contextual Markov Decision Process (CMDP) is defined by $(C, S, O, \phi, A, R, \rho, \gamma, M(c))$. $C$ is the set of contexts, $S$ is the underlying state space, $O$ and $A$ are sets of observations and actions, $\phi : S \rightarrow O$ is the observation function, $R : S \rightarrow \mathbb{R}$ is a reward-function specifying a *task*, $\gamma$ is a discount factor, and $\rho$ is the initial state distribution [35]. $M$ is a function that maps a context $c \in C$ to a Partially Observable Markov Decision Process (POMDP) [2] $M(c) = (S, O, A, R, \rho, \gamma, P^c)$ with a context-dependent transition function $P^c : S \times A \times C \rightarrow \Delta(S)$. A Markov policy $\pi : S \rightarrow \Delta(A)$ is optimal in context $c$ for reward function $R$ if it maximises the expected discounted future reward *i.e.* $\pi^*_{c,R} = \arg\max_\pi \mathbb{E}[\gamma^t R(s_{t+1})|s_0, a_0, \pi, c]$, where $\mathbb{E}[\cdot|s_0, a_0, \pi, c]$ is the expectation under state-action sequence $(s_t, a_t)_{t \geq 0}$ starting at $(s_0, a_0)$ with $s_t \sim P^c(\cdot|s_{t-1}, a_{t-1})$ and $a_t \sim \pi(\cdot|s_t)$. Note that the context $c \in C$ cannot be observed directly.

**Problem setting.** We split the CMDP into a set of training contexts $C_{\text{train}}$ and testing contexts $C_{\text{test}}$. We assume access to a dataset $\mathcal{D}_{\text{train}}$ of *unlabelled* observation-action trajectories $\tau = (o_0, a_0, o_1, \ldots, o_T)$ collected from the training contexts by a highly exploratory behaviour policy. Our goal is to pre-train an adaptive policy $\pi(a|h, z)$, where $h \in \mathbb{R}^m$ is a hidden state summarising both the context $c$ and inferred Markov state $s$, and $z \in \mathbb{R}^d$ denotes a compact representation of the task. We will pre-train this policy solely from offline data $\mathcal{D}_{\text{train}}$, without online interactions.

We will evaluate the policy on an unseen test task $R_{\text{test}}$ in an unseen test context $c_{\text{test}} \in C_{\text{test}}$. The test task is revealed either via $\mathcal{D}_{\text{test}}$, a small dataset of *labelled* observation trajectories $((o_{t-L}, \ldots, o_t), R_{\text{test}}(s_t))$ of length $L$, or as an explicit function $o \mapsto R_{\text{test}}(s)$ (like 1 at a goal state and 0 elsewhere)[1]. Unless, the agent can infer $c_{\text{test}}$ from $\mathcal{D}_{\text{test}}$, it will need to infer it from the observation-action history it observes during evaluation. This problem setting is directly equivalent to [97]'s zero-shot RL setting with a change in the environment dynamics between training and testing. As a result, we call it **zero-shot RL under changed dynamics**.

**Foundation policies.** Foundation policies (FPs) approximate the (universal) successor features [6, 11] of near-optimal policies for any task in an environment. They require access to a feature map $\varphi : S \mapsto \mathbb{R}^d$ that embeds states into a representation space in which the reward is assumed to be linear *i.e.* $R(s) = \varphi(s)^\top z$ with *weights* $z \in \mathbb{R}^d$ representing a task. The USFs $\psi : S \times A \times \mathbb{R}^d \rightarrow \mathbb{R}^d$

---

[1]Note that the agent only sees the observation, but the reward is a function of the underlying state.

are defined as the discounted sum of future features subject to a task-conditioned policy $\pi(s, z)$:

$$\psi(s_0, a_0, z) = \mathbb{E}\left[\sum_{t \geq 0} \gamma^t \varphi(s_{t+1}) | s_0, a_0, \pi(s, z)\right] \quad \forall\, s_0 \in S, a_0 \in A, z \in \mathbb{R}^d. \tag{1}$$

where the policy is trained in an actor-critic formulation [51] such that

$$\pi(s, z) \approx \arg\max_a \psi(s, a, z)^\top z, \,\forall\, s \in S, a \in A, z \in \mathbb{R}^d, \tag{2}$$

where $\psi(s, a, z)^\top z$ is the $Q$ function (critic) formed by $\psi$. During training, candidate task weights are sampled from $\mathcal{Z}$, a prior over the task space[2]. During evaluation, the test task weights are found by regressing labelled states onto the features: $z_{\text{test}} := \arg\min_z \mathbb{E}_{s \sim d}[(R_{\text{test}}(s) - \varphi(s)^\top z)^2]$, before being passed to the policy. The features can be learned with Hilbert representations [76], laplacian eigenfunctions [97], contrastive methods [97], or in service of *successor measure* prediction [10], as is the case for the forward Backward (FB) foundation policy [96] used in this work.

## 3 METHOD

Recall that our goal is to pre-train an adaptive policy $\pi(a|h, z)$ that is conditioned on $h$, a hidden state summarising both the context $c$ and inferred Markov state $s$, and task $z$. As we outlined in Section 2, the FP framework provides a principled way of pre-training $\pi(a|s, z)$ *i.e.* a policy conditioned solely on the task and Markov state. In this section, we will discuss amendments to the FP framework that allow policies to be conditioned on $h$ rather than $s$.

### 3.1 MEMORY MODELS

Following past work on RL in CMDPs, we assume that we can produce an estimate of the dynamics context $c$ and Markov state $s$ from a *trajectory* of observation-action pairs $\tau = x_0, \ldots, x_L$, where $x_n = \epsilon(o_n, a_n)$ is some encoding of an observation-action pair and $L$ is the *context length* [31, 65]. We seek a model of the form

$$y_j, h_j = f(x_j, h_{j-1}), \quad j \in [1, \ldots, L], \tag{3}$$

where $x_j, y_j$ are the inputs and outputs at time $j$, and $f$ updates a hidden state $h \in \mathbb{R}^m$ summarising the current Markov state and dynamics context prediction. This is the standard setup of a *memory model* in RL [4, 68, 69, 70, 82, 66, 73, 40, 86, 100, 8, 109], because the asymptotic inference time complexity is $\mathcal{O}(1)$ which is helpful for fast data collection, or high-frequency motor control [62]. Until recently, only Recurrent Neural Networks (RNNs) [24, 41, 17] have had this property, but newly proposed structured state-space models (S4) [32, 33, 34] and fast Transformers [98, 19, 48] have runtime complexity approaching that of RNNs, and model histories with large $L$ more accurately. We explore all of these memory models in Section 4.

### 3.2 FOUNDATION POLICIES WITH MEMORY

Equipped with memory model $f$, we now condition the FP's actor and critic on the hidden state it produces. We define *contextual* USFs as the discounted sum of future features extracted from the hidden state, subject to a policy conditioned on the inferred Markov state and dynamics context $\pi(h, z)$

$$\psi(h, z) = \mathbb{E}\left[\sum_{t \geq 0} \gamma^t \varphi(h_{t+1}) | h_0, \pi(h, z)\right] \quad h_0 = \mathbf{0}^m, \forall\, z \in \mathbb{R}^d, \tag{4}$$

where $h_t = f(x_t, h_{t-1})$ from Equation 3, $x_t$ is zero-padded for all $t < L$, and $h_0 = \mathbf{0}^m$ is an initial hidden-state of zeroes. The policy is trained such that

$$\pi(h, z) \approx \arg\max_a \psi(h, z)^\top z, \quad \forall\, h \in \mathbb{R}^m, z \in \mathbb{R}^d, \tag{5}$$

---

[2]See Appendix B.1.1 for more detail on $\mathcal{Z}$.

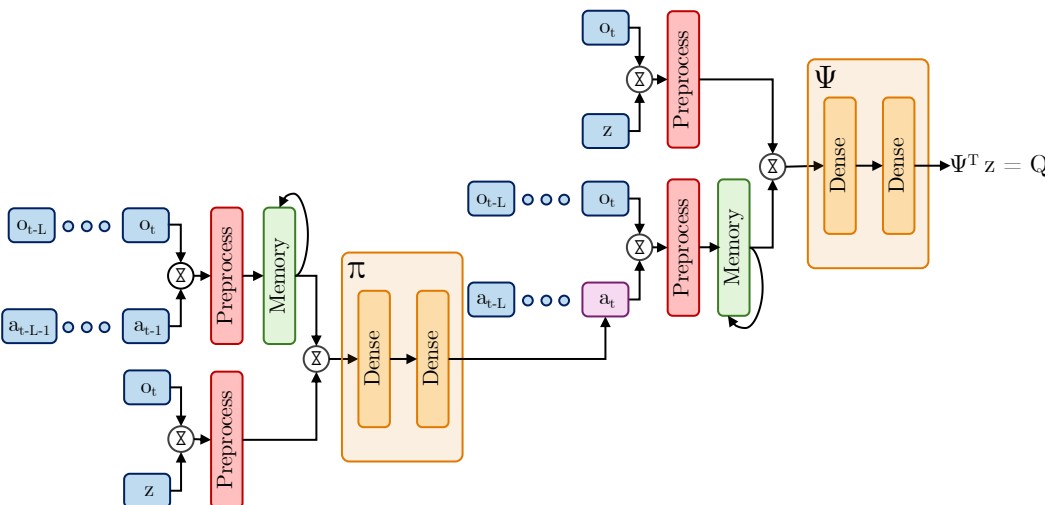

Figure 2: **Foundation policies with memory**. FPs are optimised in a standard actor critic setup [51]. The policy $\pi$ selects an action $a_t$ conditioned on a *history* of observations and actions $o_{t-L}, a_{t-L-1}, \ldots, o_t, a_{t-1}$ of length $L$ encoded by the actor's memory model, and the task vector $z$. The $Q$ function formed by the USF $\psi$ evaluates the sequence of observations and actions $o_{t-L}, a_{t-L}, \ldots, o_t, a_t$ encoded by the critic's memory model for task $z$. The architecture of an FP *without* memory is illustrated in Figure 6 in Appendix B for comparison.

where $\psi(h,z)^\top z$ is the $Q$ function (critic) formed by $\psi$. Training proceeds exactly as with conventional USFs, and the test-time task weights are found by regressing labelled states onto the hidden-state features: $z_{\text{test}} := \arg\min_z \mathbb{E}_{s_t, (o_{t-L:t}, a_{t-L:t}) \sim d}[(R_{\text{test}}(s_t) - \varphi(f((o_{t-L:t}, a_{t-L:t}), h_{0:L})^\top z)^2]$, before being passed to the policy. The full architecture and optimisation procedure is summarised in Figure 2. We found that using a shared memory model for the actor and critic led to model collapse, so use separate memory models for each. This corroborates the findings of [73]. Full implementation details are provided in Appendix B. In the experiments discussed in Section 4 we use FB representations as our FP which follow a slightly different training procedure. We discuss these details in Appendix B.

## 4 EXPERIMENTS

In this section we perform an empirical study to evaluate our proposed method. We seek answers to three questions: (**Q1**) Can our method encode trajectories into a Markov state for use in solving one task in an environment? (**Q2**) Can our method generalise to unseen tasks in an environment with different dynamics to those seen in training? I.e. can our method perform *zero-shot RL under changed dynamics*? And (**Q3**) Can our method generalise to unseen tasks in a completely different environment to those seen in training? I.e. can our method perform *zero-shot environment generalisation*?

### 4.1 SETUP

**Environments.** We respond to **Q1** using the POPGym benchmark [68], a set of tests that evaluate an agent's ability to infer Markov states from trajectories of observations and actions. We only evaluate on the "Hard" versions of CartPole, Pendulum, Noisy CartPole, Noisy Pendulum and Repeat Previous environments following [62]. For these experiments, we allow the agent to recondition its policy on the previous $L$ observation-action pairs every step so we can disentangle the memory model's ability to accurately model the Markov state from its ability to carry forward an accurate hidden state. For all other experiments we do not allow such re-conditioning and require the policy to condition on only the previous hidden state, current observation-action pair, and task. Note for

these experiments $C_{\text{train}} = C_{\text{test}}$, so we are not yet testing whether our method can generalise across contexts.

We respond to **Q2** using the ExORL benchmark [108], a set of tests that evaluate an agent's ability to generalise to unseen tasks on the DeepMind Control Suite [93]. We evaluate on the same environments as [97]: Walker, Maze, Cheetah and Quadruped, removing the velocities from each of the state spaces to ensure the observations are not Markov, and call these variants *occluded*. To evaluate dynamics generalisation we train on datasets collected from environment instances where the robot's mass and damping coefficient are scaled to $\{0.5x, 1.5x\}$ of their usual values. We then evaluate on environment instances where the robot's mass and damping coefficient are scaled by $\{1.0x, 2.0x\}$ of their usual values, where $1.0x$ requires the agent to generalise via interpolation, and $2.0x$ requires the agent to generalise via extrapolation [74]. We evaluate on all tasks provided by the DeepMind control suite, and increase the number of goals in Maze from 4 to 20 for a total of 32 tasks across 4 environments.

We also respond to **Q3** using the ExORL benchmark [108]. This time we train on Walker-Occluded and Quadruped-Occluded and test on Cheetah-Occluded. The dynamics are unscaled (i.e. $1.0x$) and, as before, we evaluate on all tasks provided by the DeepMind control suite.

Aggregation across tasks or environments is always summarised by the Interquartile Mean (IQM) and standard deviation following the recommendations of [1]. On POPGym we report the mean-max epoch reward (MMER) metric used in the original paper. On ExORL, we report scores from the learning step for which the all-task IQM is maximised across seeds. Full experimental details are provided Appendix A, and a full description of our evaluation protocol is provided in Appendix A.3.

**Baselines.** We use FB [96] and HILP [76] as our FP baselines. FB is the most performant FP utilising successor measures, and HILP is the most performant FP utilising successor features. Both methods assume access to the Markov state for training as discussed in Section 2. So, instead of conditioning their predictions on a single observation, we provide them a stack of the 4 most recent observations i.e. $s_t = (o_{t-3}, o_{t-2}, o_{t-1}, o_t)$, also known as *frame-stacking* [67]. Frame-stacking is a naive method for inferring a Markov state from a short trajectory of observations, and is the first solution one would use when faced with our problem. We also baseline against a single-task, memory-based, supervised RL method. For this we use Offline TD3 [28] with a GRU memory model [17], which we refer to as **TD3-GRU**. Offline TD3 is the most performant single-task method on the ExORL benchmark [108]; TD3-GRU is the most performant method in [73], and TD3 with an LSTM memory model was shown to be particularly performant in [66]. TD3-GRU should indicate how well an agent optimising for one task performs if provided reward supervision.

**Datasets.** Though FPs can be deployed online they require an exploration policy for data collection. To disentangle test-time performance from an agent's data collection ability, we collect datasets on their behalf in advance using RND [11], an unsupervised RL algorithm. Agents trained on datasets collected with RND exhibit better performance than comparable methods like APS [60], APT [61], Proto [107] and DIAYN [25] in [108, 97, 76]. RND is run for 5 million learning steps in each of our environments and every transition is cached. For the supervised baseline TD3-GRU, transitions are relabelled with the appropriate rewards for a given task following [108]. All other methods are trained on these datasets reward-free.

**Memory models.** We test the performance of FPs equipped with three memory models. We use the most performant versions from each of the categories discussed in Section 3: attention-based, state-space based, and RNN-based. For our attention-based memory model we use a Transformer [98] with *FlashAttention* [19] for faster inference than a conventional Transformer. For our state-space-based memory model we use Diagonalized S4 [33], which uses a diagonal update matrix to perform faster training and inference than the popular S4 model [32]. For our RNN-based memory model we use a GRU [17] as it is the most performant RNN on the POPGym benchmark. Hereafter we refer to the FB models we augment with these as **FB-TF**, **FB-S4** and **FB-GRU** respectively. To ensure a fair comparison across memory models, we follow [68] and restrict each model to a fixed hidden state size, rather than a fixed parameter count. Concretely, we allow each model a hidden state size of $32^2 = 1024$ dimensions. In Section 4.3 we condition the models on trajectories of length 32, so a hidden state size of $32^2$ allows the attention-based, and state-space models to perform their tensor products across the full input trajectory, and gives the RNN two 512-dimension

layers in which to summarise the trajectory. Full implementation details are provided in Appendix B.

## 4.2 POPGYM

We report the aggregate performance of all FB-based algorithms on the POPGym environments in Figure 3. Our supervised baseline, TD3-GRU, performs similarly to the PPO-GRU approach in the original POPGym paper. FB with frame-stacking performs poorly, reaching only 30% of TD3-GRU's aggregate score. Our three memory-based methods perform comparitively better, with FB-TF reaching 80% of TD3-GRU's performance, and FB-S4 and FB-GRU matching TD3-GRU's performance. We find that all methods fail on the RepeatPreviousHard environment, where other in-context RL agents have shown strong performance [62, 31]. This task requires the agent to remember the suit of a card dealt 64 timesteps ago (Appendix A), and our models are trained with context length $L = 64$. This suggests that the memory models are not accurately recalling information from the start of their context. The implications of our choice of length $L$ are discussed in Section 5.

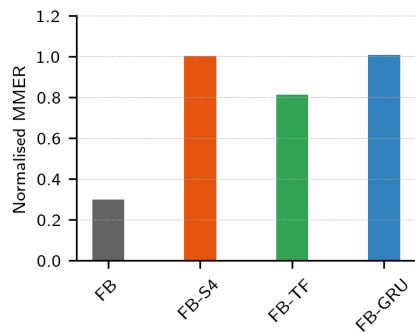

Figure 3: **POPGym results.** Aggregate mean-maximum epoch reward (MMER) across all POPGym environments, normalised w.r.t. TD3-GRU performance.

## 4.3 ZERO-SHOT RL UNDER CHANGED DYNAMICS

We report the aggregate performance of all algorithms on our *zero-shot RL under changed dynamics* experiments in Figure 4 *(left)*, and the ratios of interpolation/extrapolation performance to train performance in Figure 4 *(right)*. As with our POPGym experiments, FB performs poorly, reaching $\sim 25\%$ of the performance of our supervised baseline on the training environments. HILP performs slightly better, as we would expect given its results on ExORL in [76], but still much poorer than TD3-GRU. Of the three FPs with memory, FB-GRU performs best on train, interpolation and extrapolation evaluations, with results relative to the supervised baseline similar to FB trained on Markov states in [97]. Aggregate test performance approximately matches TD3-GRU aggregate test performance despite not seeing rewards during training. FB-TF exhibits the best interpolation-to-train ratio, and FB-GRU the best extrapolation-to-train ratio.

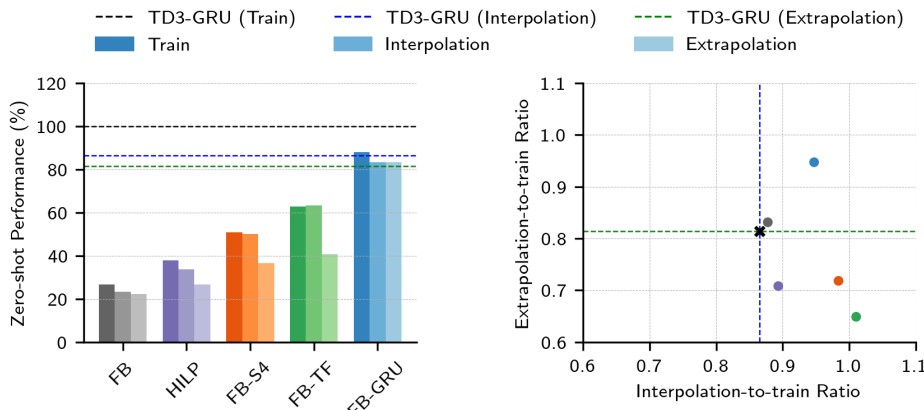

Figure 4: **Zero-shot dynamics generalisation on ExORL.** *(Left)* Aggregate performance across all ExORL tasks and domains normalised w.r.t. TD3-GRU performance, averaged over 5 seeds. We train on dynamics $\{0.5x, 1.5x\}$ their typical values and evaluate on $1.0x$ (interpolation) and $2.0x$ (extrapolation). *(Right)* The ratios of interpolation and extrapolation performance to train performance.

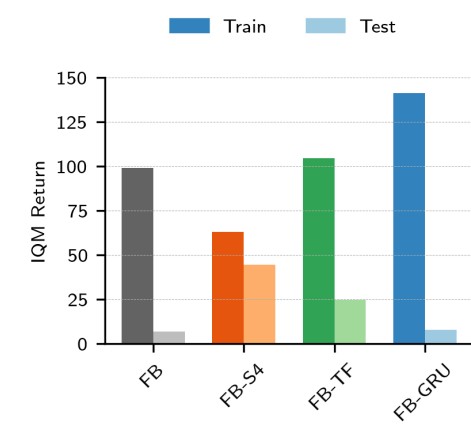

Figure 5: **Zero-shot environment generalisation on ExORL.** Aggregate performance across all tasks in the train environments (Walker, Quadruped), and the test environment (Cheetah), averaged across 5 seeds.

### 4.4 ZERO-SHOT ENVIRONMENT GENERALISATION

We report the aggregate performance of all FB-based algorithms on our *zero-shot environment generalisation experiments* in Figure 5. Here, FB-GRU performs best on the training environments, but poorest on the testing environment, with FB performing similarly poorly on the testing environment. FB-S4 improves performance on the testing environment by $\sim$ 4x over FB, but at the cost of reducing training environment performance by half. FB-TF improves both training performance by 5% over FB and triples test performance. We emphasise that, although FPs with memory do improve zero-shot environment generalisation performance in some cases, the absolute returns remain low (a max of 33 for FB-S4 out of a possible 1000) suggesting their is significant room for improvement.

## 5 DISCUSSION AND LIMITATIONS

**Context length.** In Section 4 we train agents with a context length $L = 64$ timesteps, which is the maximum context length we could afford with our computational budget[3]. We see two limitations with this. First, we have assumed the dynamics context and task for all of our experiments can be inferred from this context, but it is not clear that this is the case. Indeed, TD3-GRU with reward supervision and a maximally exploratory dataset does not match its performance with Markov states from [97]. Second, successful episodes run for a minimum of 200 timesteps (as in PendulumHard) and a maximum of 1000 timesteps (as in ExORL), meaning we never train the memory model over full episodes, nor can we reliably maintain an episode's full trajectory in context at test-time. This introduces situations where the hidden state will be erroneously initialised mid-episode during training, creating well-established theoretical issues for memory models in RL [68], though these are yet to prove critical empirically [73, 66].

The obvious solution to these problems, were it available to us, would be to increase $L$ until it is the maximum episode length, and train for longer as in [31, 62, 68]. However, even if we were to do this, any real-world deployment may induce episodes longer than this assumed max length, or indeed we may wish to operate in the non-episodic, continual setting. The existing literature implicitly assumes that if $L$ is very large such issues will resolve themselves, but this is not clear to us. Exploring how to deal with such situations is an important future research direction.

**Datasets.** As outlined in Section 4.1, we train all methods on datasets pre-collected with RND [11] which is a highly exploratory algorithm designed for maximising data heterogeneity. However, deploying such an algorithm in any real setting may be costly, time-consuming or dangerous. As a result, our proposals are more likely to be trained on real-world datasets that are smaller and more

---

[3]Our shared resource limits us to a maximum run length of 24 hours per GPU, and the ExORL runs took approximately 20 hours on one A100. See Appendix A.4 for more detail.

homogeneous. It is not clear how our specific proposals will interact with such datasets. If, for example, the dataset only exhibits parts of the state space from which the dynamics cannot be well-inferred, like a robot stuck stationary, then we would expect our proposals to struggle. Indeed, with poor coverage of the state-action space, we would expect to see the OOD pathologies seen in the single-task Offline RL setting [52, 59]. That said, the proposals of [45] for conducting zero-shot RL from real-world datasets could be integrated into our proposals easily, and may help.

## 6 RELATED WORK

**Generalist policy pre-training** FPs build upon successor representations [20], universal value function approximators [85], successor features [6] and successor measures [10]. The state-of-the-art methods instantiate these ideas as either universal successor features (USFs) [11] or forward-backward (FB) representations [96, 97], with recent work showing they can be trained on low quality datasets [45], or used to perform a range of imitation learning techniques efficiently [80]. A representation learning method is required to learn the features for USFs, with past works using inverse curiosity modules [79], diversity methods [60, 38], Laplacian eigenfunctions [101], or contrastive learning [16]. No works have yet explored the generalisation capacity of FPs to unseen dynamics. Two concurrent lines of work on *goal-conditioned* RL and *in-context* RL also seek to build generalist policies. Goal-conditioned RL methods train policies to reach any goal state from any other goal state [75, 63, 104, 26, 99], but lack the ability to generalise to tasks with "dense" reward functions, like those on the locomotion tasks in ExORL. In-context RL methods train policies using sequence models conditioned on reward-labelled histories [14, 43, 58, 83, 110, 13, 30, 88, 103, 102, 31, 62, 94, 23], but, unlike FPs, do not have a robust mechanism for training without access to rewards.

**Dynamics Generalisation** Dynamics generalisation in RL is a well-established problem [50, 65, 74]. Common remedies include: data augmentation [81, 5, 106, 37, 36, 55], domain randomisation [95, 21, 46, 47, 77], learning context-aware policies [87, 57, 9, 44], and meta-learning [15, 83, 27, 71, 72]. Our work is most similar to those that tackle dynamics generalisation by conditioning policies on dynamics inferred with a memory model [73, 18]. Where these methods are concerned with generalising to one unseen task in unseen dynamics contexts, our method can generalise to more than one unseen tasks in unseen dynamics contexts.

## 7 CONCLUSION

In this paper, we explored augmenting Foundation Policies (FPs) with memory models to allow them to condition policies on a dynamics context inferred from a history of observations and actions. We evaluated our proposals with attention, state-space, and RNN-based memory models on POPGym, a memory benchmark, and ExORL, an unsupervised RL benchmark. Our results show that GRUs achieve the best generalisation to unseen tasks and dynamics for a given recurrent state size, approaching the performance of a supervised baseline that has access to task information during training and significantly outperforming memory-free FPs. We believe our proposals represent a further step toward the development of generalist, adaptive agents.

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

# APPENDICES

## A EXPERIMENTAL DETAILS

### A.1 POPGYM

We consider 5 environments from the POPGym benchmark [68] which is built atop the OpenAI gym [12]. Each tests the agents ability to summarise a trajectory of observations and actions into a Markov state for use in solving one downstream task. Following [62] we only consider the *hard* variants, because the other variants are considered too straightforward.

**Stateless CartPole Hard.** The cartpole environment from [7], but with the angular and linear positions removed from the observation. The agent must integrate to compute positions from velocity and balance the pole atop the cart to receive reward. The *hard* variant requires the pole to be balanced for 600 timesteps (the *easy* and *medium* variants require the pole to be balanced for fewer timesteps).

**Noisy Stateless CartPole Hard.** The same as Stateless CartPole Hard but with Gaussian noise added to observations. The *hard* variant sets the standard deviation of the noise $\sigma = 0.3$ (the *easy* and *medium* set $\sigma = 0.1$ and $\sigma = 0.2$ respectively).

**Stateless Pendulum Hard.** The swing-up pendulum [22] with the angular position information removed. The agent must integrate to compute positions from velocity and swing the pendulum up to receive reward. The *hard* variant requires the pendulum to be balanced for 200 timesteps (the *easy* and *medium* variants require the pole to be balanced for fewer timesteps).

**Noisy Stateless Pendulum Hard.** The same as Stateless Pendulum Hard but with Gaussian noise added to observations. The *hard* variant sets the standard deviation of the noise $\sigma = 0.3$ (the *easy* and *medium* set $\sigma = 0.1$ and $\sigma = 0.2$ respectively).

**Repeat Previous Hard.** Observations contain one of four values. The agent is rewarded for outputting the observation from some constant k timesteps ago, i.e. observation $o_{t-k}$ at time $t$. The *hard* variant sets $k = 64$ (the *easy* and *medium* variants set $k < 64$).

### A.2 EXORL

We consider 4 environments (three locomotion and one goal-directed) from the ExORL benchmark [108] which is built atop the DeepMind Control Suite [93]. We occlude their states by removing all velocity components, similar to [73, 66]. Environments are visualised here: https://www.youtube.com/watch?v=rAai4QzcYbs. The domains are summarised in Table 1.

**Walker-Occluded.** A two-legged robot required to perform locomotion starting from bent-kneed position. The observation and action spaces are 17 and 6-dimensional respectively (after occlusion), consisting of joint torques and positions. ExORL provides 4 tasks `stand`, `walk`, `run` and `flip`. The reward function for `stand` motivates straightened legs and an upright torso; `walk` and `run` are supersets of `stand` including reward for small and large degrees of forward velocity; and `flip` motivates angular velocity of the torso after standing. Rewards are dense.

**Quadruped-Occluded.** A four-legged robot required to perform locomotion inside a 3D maze. The observation and action spaces are 67 and 12-dimensional respectively (after occlusion), consisting of joint torques and positions. We evaluate on 4 tasks `stand`, `run`, `walk` and `jump`. The reward function for `stand` motivates a minimum torso height and straightened legs; `walk` and `run` are supersets of `stand` including reward for small and large degrees of forward velocity; and `jump` adds a term motivating vertical displacement to stand. Rewards are dense.

**Maze-Occluded.** A 2D maze with four rooms where the task is to move a point-mass to one of the rooms. The observation and action spaces are both 2-dimensional (after occlusion); the observation space consists of $x, y$ positions of the mass, the action space is the $x, y$ tilt angle. ExORL provides four reaching tasks `top left`, `top right`, `bottom left` and `bottom right` corresponding to each room. We add four other goals in each room following [97] to provide a total of 20 goal reaching tasks. The mass is always initialised in the top left and the reward is proportional to the distance from the goal, though is sparse i.e. it only registers once the agent is reasonably close to the goal.

**Cheetah-Occluded.** A running two-legged robot. The observation and action spaces are 10 and 6-dimensional respectively (after occlusion), consisting of positions of robot joints. We evaluate on 4 tasks: `walk`, `walk backward`, `run` and `run backward`. Rewards are linearly proportional either a forward or backward velocity–2 m/s for walk and 10 m/s for run.

## A.3 EVALUATION PROTOCOL

We evaluate the cumulative reward (hereafter called score) achieved by all methods across three seeds in POPGym and 5 seeds in ExORL. We report task scores as per the best practice recommendations of [1]. Concretely, we run each algorithm for 500k learning steps (1m for ExORL), evaluating task scores at checkpoints of 20,000 steps. At each checkpoint, we perform 10 rollouts, record the score of each, and find the interquartile mean (IQM). We average across seeds at each checkpoint. We extract task scores from the learning step for which the all-task IQM is maximised across seeds. Results are reported with their associated standard deviation. Aggregation across tasks, domains and datasets is always performed by evaluating the IQM.

## A.4 COMPUTATIONAL RESOURCES

We train our models on NVIDIA A100 GPUs. Training TD3-GRU to solve one task on one GPU takes approximately 6 hours for POPGym and 8 hours for ExORL. One run of FB-stack and SF-stack on one domain (for all tasks) takes approximately 3 hours for POPGym and 5 hours for ExORL on one GPU. One run of the memory-based FPs on one domain (for all tasks) on one GPU in approximately 20 hours. Note the POPGym experiments run for 500k learning steps, whereas the ExORL experiments run for 1m learning steps. As a result, our core experiments on the ExORL benchmark used approximately 65 GPU days of compute.

# B IMPLEMENTATION DETAILS

## B.1 FOUNDATION POLICIES

FB and HILP follow the implementations by [76] which follow [97], other than the batch size which we reduce from 1024 to 512 to reduce the computational expense of each run without limiting performance as per [45]. Hyperparameters are reported in Table 2. An illustration of a standard FP architecture is provided in Figure 6, for comparison with the FP with memory architecture in Figure 2.

**Forward Representation** $F(o, a, z)$ **/ USF** $\psi(o, a, z)$. Inputs $(o_{t-L:t}, a)$ and state-task pairs $(o, z)$ are preprocessed by feedforward MLPs that embed their inputs into a 512-dimensional space. These embeddings are concatenated and passed through a third feedforward MLP $F$ / $\psi$ which outputs a $d$-dimensional embedding vector. The Transformer memory model with Flash Attention follows the exact implementation in [31]; the S4d memory model follows the exact implementation in [68], and the GRU memory model follows the exact implementation provided by Torch.

Table 1: **ExORL domain summary.** *Dimensionality* refers to the relative size of state and action spaces. *Type* is the task categorisation, either locomotion (satisfy a prescribed behaviour until the episode ends) or goal-reaching (achieve a specific task to terminate the episode). *Reward* is the frequency with which non-zero rewards are provided, where dense refers to non-zero rewards at every timestep and sparse refers to non-zero rewards only at positions close to the goal. Green and red colours reflect the relative difficulty of these settings.

| Environment | Dimensionality | Type | Reward |
|---|---|---|---|
| Walker-Occluded | Low | Locomotion | Dense |
| Quadruped-Occluded | High | Locomotion | Dense |
| Maze-Occluded | Low | Goal-reaching | Sparse |
| Cheetah-Occluded | Low | Locomotion | Dense |

Table 2: **FP Hyperparameters.**

| Hyperparameter | Value |
|---|---|
| Latent dimension $d$ | 50 |
| $F$ / $\psi$ dimensions | (1024, 1024) |
| $B$ dimensions | (512, 512) |
| Preprocessor dimensions | (512, 512) |
| Transformer heads | 4 |
| Transformer / S4d model dimension | 32 |
| GRU dimensions | (512, 512) |
| Context length $L$ | 32 (Section 4.3), 64 (Sections 4.2 and 4.4) |
| Frame stacking (FB & HILP) | 4 |
| Std. deviation for policy smoothing $\sigma$ | 0.2 |
| Truncation level for policy smoothing | 0.3 |
| Learning steps | 1,000,000 (ExORL), 500,000 (POPGym) |
| Batch size | 512 |
| Optimiser | Adam [49] |
| Learning rate | 0.0001 |
| Discount $\gamma$ | 0.98 |
| Activations (unless otherwise stated) | ReLU |
| Target network Polyak smoothing coefficient | 0.01 |
| $z$-inference labels | 10,000 |
| $z$ mixing ratio | 0.5 |
| HILP representation discount factor | 0.98 |
| HILP representation expectile | 0.5 |
| HILP representation target smoothing coefficient | 0.005 |

**Backward Representation** $B(o_{t-L:t})$ **(for FB).** Inputs are preprocessed by feedforward MLPs that embed their inputs into a 512-dimensional space then passed to the backward representation $B$ which is a feedforward MLP that outputs a $d$-dimensional embedding vector.

**Actor** $\pi(o_{t-L:t}, z)$. Inputs $(o_{t-L:t}, a)$ and state-task pairs $(o, z)$ are preprocessed by feedforward MLPs that embed their inputs into a 512-dimensional space. These embeddings are concatenated and passed through a third feedforward MLP which outputs a $a$-dimensional vector, where $a$ is the action-space dimensionality. A `Tanh` activation is used on the last layer to normalise their scale. As per [29]'s recommendations, the policy is smoothed by adding Gaussian noise $\sigma$ to the actions during training.

**Misc.** Layer normalisation [3] and `Tanh` activations are used in the first layer of all MLPs to standardise the inputs.

### B.1.1 $z$ SAMPLING

FPs require a method for sampling the task vector $z$ at each learning step. [97] employ a mix of two methods, which we replicate:

1. Uniform sampling of $z$ on the hypersphere surface of radius $\sqrt{d}$ around the origin of $\mathbb{R}^d$,
2. Biased sampling of $z$ by passing states $s \sim \mathcal{D}$ through the backward representation $z = B(s)$. This also yields vectors on the hypersphere surface due to the $L2$ normalisation described above, but the distribution is non-uniform.

We sample $z$ 50:50 from these methods at each learning step.

### B.2 TD3-GRU

We adopt the same implementation and hyperparameters as is used on the ExORL benchmark. Hyperparameters are reported in Table 3. The memory module follows the implementation from [73] and uses a seperate encoder for the actor and critic.

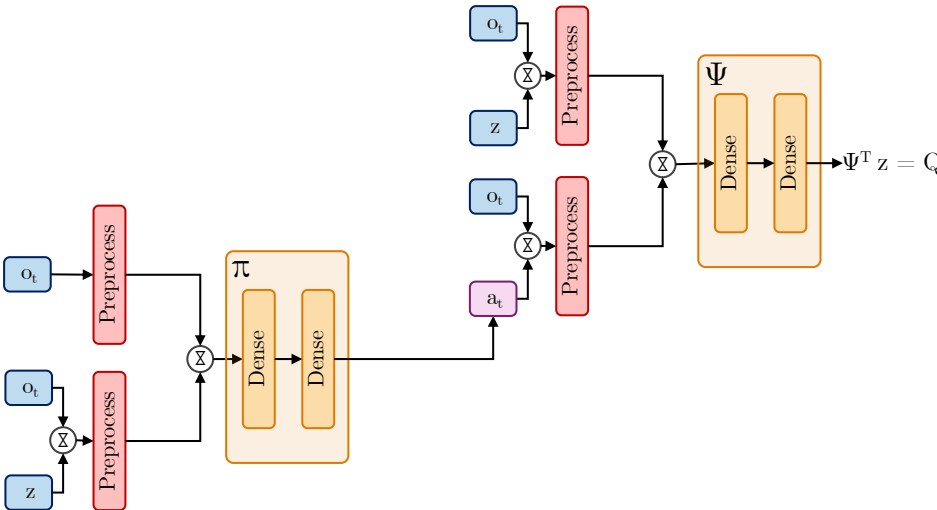

Figure 6: **Foundation policies *without* memory**. FPs are optimised in a standard actor critic setup [51]. The policy $\pi$ selects an action $a_t$ conditioned on a the current observation $o_t$, and the task vector $z$. The $Q$ function formed by the USF $\psi$ evaluates the action $a_t$ given the current observation $o_t$ and task $z$.

**Critic(s).** TD3 employs double Q networks, where the target network is updated with Polyak averaging via a momentum coefficient. The critics are feedforward MLPs that take a state-action pair $(s, a)$ as input and output a value $\in \mathbb{R}^1$.

**Actor.** The actor is a standard feedforward MLP taking the state $s$ as input and outputting an $a$-dimensional vector, where $a$ is the action-space dimensionality. The policy is smoothed by adding Gaussian noise $\sigma$ to the actions during training.

**Misc.** As is usual with TD3, layer normalisation [3] is applied to the inputs of all networks.

Table 3: **TD3-GRU hyperparameters.**

| Hyperparameter | Value |
|---|---|
| Critic dimensions | (1024, 1024) |
| Actor dimensions | (1024, 1024) |
| GRU dimensions | (512, 512) |
| Preprocessor dimensions | (512, 512) |
| Learning steps | 1,000,000 (ExORL), 500,000 (POPGym) |
| Batch size | 512 |
| Optimiser | Adam |
| Learning rate | 0.0001 |
| Discount $\gamma$ | 0.98 |
| Activations | ReLU |
| Target network Polyak smoothing coefficient | 0.01 |
| Std. deviation for policy smoothing $\sigma$ | 0.2 |
| Truncation level for policy smoothing | 0.3 |

### B.3 Code References

This work was enabled by: Python [84], NumPy [39], PyTorch [78], Pandas [64] and Matplotlib [42].

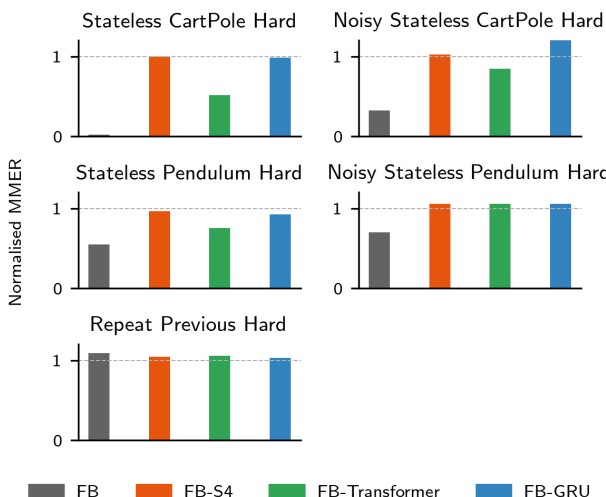

Figure 7: **Per-environment POPGym results.** The results are aggregated over 3 seeds, visualised by environment, and report the normalised MMER as with Table 4.

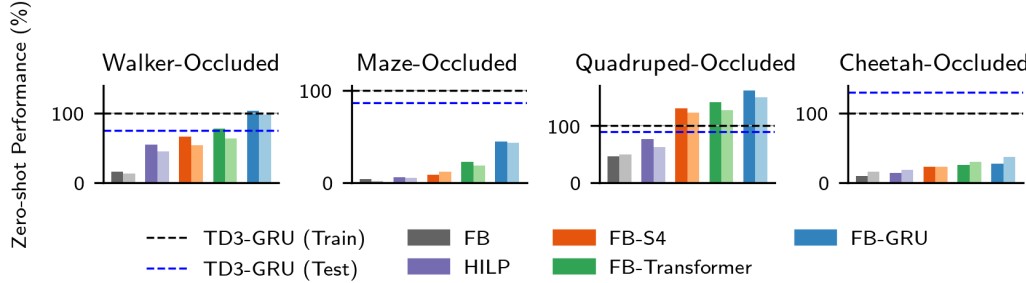

Figure 8: **Per-environment zero-shot dynamics generalisation results.** The results are aggregated over 5 seeds and all tasks in each environment, and show the normalised IQM w.r.t. TD3-GRU.

## C    EXTENDED RESULTS

We report a full breakdown of our results summarised in Sections 4.2, 4.3, 4.4. Table 4 reports results on POPGym from Section 4.2, Table 5 reports results on the zero-shot dynamics generalisation experiments from Section 4.3, and Table 6. Additionally, Figures 7 and 8 show plots where the results are aggregated by environment, and Figure 9 show plots where the zero-shot dynamics generalisation results are aggregated by task.

Table 4: **Full results on the POPGym environments (3 seeds).** We report the *unnormalised* mean-max epoch return (MMER) return $\pm$ the standard deviation averaged over 3 seeds.

| Environment | TD3-gru | FB-stack | **FB-TF (ours)** | **FB-S4 (ours)** | **FB-GRU (ours)** |
|---|---|---|---|---|---|
| NoisyStatelessCartPoleHard | $0.156 \pm 0.011$ | $0.05 \pm 0.0$ | $0.132 \pm 0.012$ | $0.16 \pm 0.022$ | $0.196 \pm 0.021$ |
| NoisyStatelessPendulumHard | $0.543 \pm 0.004$ | $0.381 \pm 0.033$ | $0.572 \pm 0.005$ | $0.572 \pm 0.007$ | $0.572 \pm 0.009$ |
| RepeatPreviousHard | $-0.418 \pm 0.012$ | $-0.455 \pm 0.002$ | $-0.441 \pm 0.002$ | $-0.436 \pm 0.016$ | $-0.431 \pm 0.005$ |
| StatelessCartPoleHard | $1.0 \pm 0.0$ | $0.017 \pm 0.0$ | $0.515 \pm 0.259$ | $1.0 \pm 0.0$ | $0.983 \pm 0.025$ |
| StatelessPendulumHard | $0.8 \pm 0.032$ | $0.436 \pm 0.033$ | $0.601 \pm 0.015$ | $0.77 \pm 0.032$ | $0.742 \pm 0.01$ |

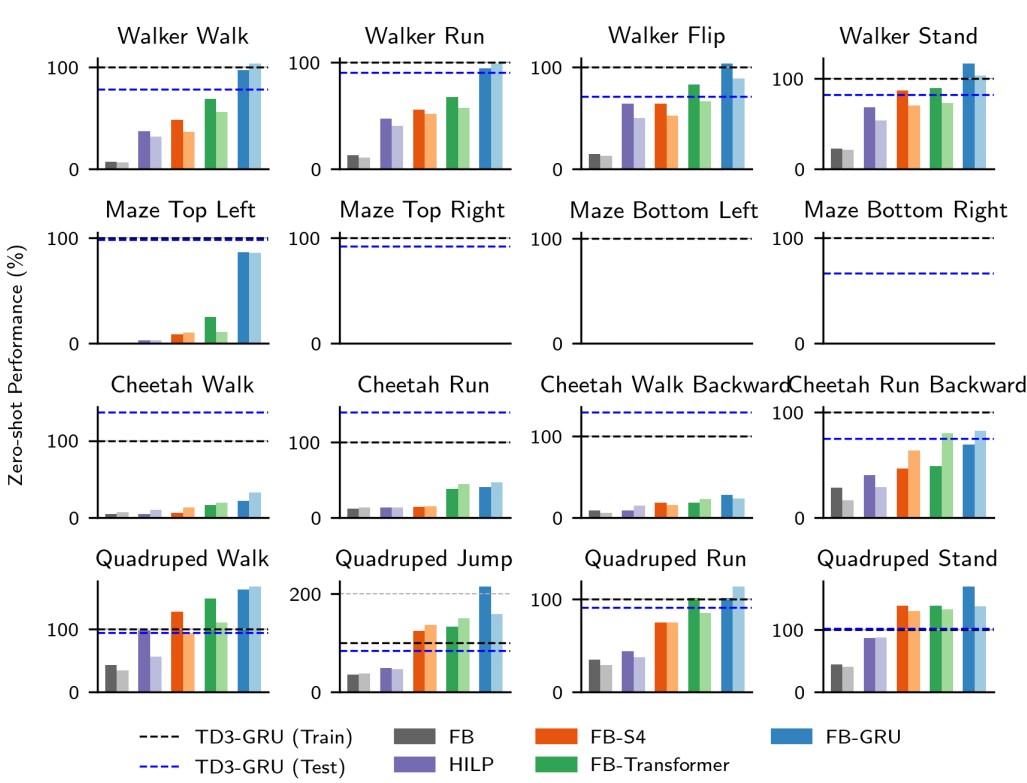

Figure 9: **Per-task zero-shot dynamics generalisation results.** The results are aggregated over 5 seeds, and show the normalised IQM w.r.t. TD3-GRU.

Table 5: **Full results on the ExORL dynamics generalisation experiments (5 seeds).** For each dataset-domain pair, we report the score at the step for which the all-task IQM is maximised when averaging across 5 seeds $\pm$ the standard deviation.

| Dynamics | Environment | Task | TD3-gru | HILP-stack | FB-stack | **FB-TF (ours)** | **FB-S4 (ours)** | **FB-GRU (ours)** |
|---|---|---|---|---|---|---|---|---|
| 0.5x | Cheetah | All tasks | $111 \pm _{98}$ | $21 \pm _6$ | $32 \pm _7$ | $25 \pm _{11}$ | $28 \pm _3$ | $43 \pm _{18}$ |
| | | Run | $26 \pm _6$ | $7 \pm _3$ | $13 \pm _2$ | $5 \pm _3$ | $10 \pm _2$ | $2 \pm _{14}$ |
| | | Run Backward | $16 \pm _9$ | $5 \pm _9$ | $5 \pm _6$ | $10 \pm _8$ | $5 \pm _4$ | $12 \pm _{16}$ |
| | | Walk | $233 \pm _{76}$ | $26 \pm _{33}$ | $77 \pm _{13}$ | $16 \pm _7$ | $53 \pm _{13}$ | $14 \pm _{67}$ |
| | | Walk Backward | $196 \pm _{122}$ | $36 \pm _{18}$ | $27 \pm _{18}$ | $64 \pm _{48}$ | $41 \pm _{18}$ | $89 \pm _{63}$ |
| | Maze | Multi goal | $413 \pm _{346}$ | $26 \pm _{20}$ | $80 \pm _{45}$ | $18 \pm _{25}$ | $20 \pm _{18}$ | $153 \pm _{40}$ |
| | Quadruped | All tasks | $279 \pm _{32}$ | $203 \pm _{31}$ | $123 \pm _{25}$ | $327 \pm _{21}$ | $541 \pm _{71}$ | $330 \pm _{179}$ |
| | | Jump | $290 \pm _{115}$ | $104 \pm _{76}$ | $85 \pm _{25}$ | $315 \pm _{86}$ | $698 \pm _{60}$ | $311 \pm _{273}$ |
| | | Run | $268 \pm _{88}$ | $104 \pm _{46}$ | $96 \pm _{73}$ | $210 \pm _{74}$ | $319 \pm _{100}$ | $178 \pm _{112}$ |
| | | Stand | $322 \pm _{83}$ | $311 \pm _{90}$ | $151 \pm _{55}$ | $469 \pm _{140}$ | $824 \pm _{108}$ | $415 \pm _{337}$ |
| | | Walk | $234 \pm _{113}$ | $270 \pm _{58}$ | $123 \pm _{69}$ | $289 \pm _{40}$ | $367 \pm _{110}$ | $362 \pm _{84}$ |
| | Walker | All tasks | $646 \pm _{224}$ | $446 \pm _{114}$ | $89 \pm _{13}$ | $451 \pm _{34}$ | $321 \pm _{24}$ | $533 \pm _{46}$ |
| | | Flip | $570 \pm _{16}$ | $409 \pm _{182}$ | $74 \pm _{25}$ | $425 \pm _{46}$ | $330 \pm _{28}$ | $489 \pm _{123}$ |
| | | Run | $249 \pm _{11}$ | $169 \pm _{26}$ | $33 \pm _3$ | $167 \pm _{10}$ | $117 \pm _8$ | $193 \pm _{10}$ |
| | | Stand | $847 \pm _{30}$ | $827 \pm _{98}$ | $182 \pm _{29}$ | $778 \pm _{15}$ | $594 \pm _{49}$ | $934 \pm _{17}$ |
| | | Walk | $723 \pm _{30}$ | $372 \pm _{196}$ | $43 \pm _{26}$ | $425 \pm _{74}$ | $243 \pm _{16}$ | $504 \pm _{66}$ |
| 1x | Cheetah | All tasks | $146 \pm _{206}$ | $36 \pm _{36}$ | $72 \pm _{24}$ | $40 \pm _{11}$ | $47 \pm _{14}$ | $54 \pm _{38}$ |
| | | Run | $37 \pm _{21}$ | $4 \pm _{11}$ | $25 \pm _{21}$ | $5 \pm _3$ | $16 \pm _7$ | $3 \pm _{24}$ |
| | | Run Backward | $11 \pm _6$ | $0 \pm _{17}$ | $6 \pm _8$ | $16 \pm _{14}$ | $11 \pm _6$ | $20 \pm _{19}$ |
| | | Walk | $524 \pm _{254}$ | $102 \pm _{116}$ | $163 \pm _{66}$ | $31 \pm _{30}$ | $92 \pm _{45}$ | $49 \pm _{152}$ |
| | | Walk Backward | $255 \pm _{192}$ | $18 \pm _{16}$ | $59 \pm _{59}$ | $92 \pm _{42}$ | $53 \pm _{19}$ | $69 \pm _{69}$ |
| | Maze | Multi goal | $354 \pm _{354}$ | $11 \pm _{18}$ | $58 \pm _{32}$ | $35 \pm _{33}$ | $21 \pm _{17}$ | $154 \pm _{47}$ |
| | Quadruped | All tasks | $232 \pm _{77}$ | $179 \pm _{19}$ | $133 \pm _{29}$ | $360 \pm _{23}$ | $445 \pm _{56}$ | $403 \pm _{160}$ |
| | | Jump | $218 \pm _{95}$ | $108 \pm _{34}$ | $135 \pm _{69}$ | $359 \pm _{58}$ | $557 \pm _{101}$ | $409 \pm _{210}$ |
| | | Run | $246 \pm _{87}$ | $76 \pm _{82}$ | $106 \pm _{77}$ | $277 \pm _{44}$ | $296 \pm _{93}$ | $203 \pm _{83}$ |
| | | Stand | $390 \pm _{130}$ | $313 \pm _{48}$ | $135 \pm _{28}$ | $565 \pm _{140}$ | $677 \pm _{125}$ | $532 \pm _{299}$ |
| | | Walk | $192 \pm _{103}$ | $168 \pm _{49}$ | $88 \pm _{59}$ | $248 \pm _{47}$ | $286 \pm _{54}$ | $362 \pm _{135}$ |
| | Walker | All tasks | $519 \pm _{192}$ | $385 \pm _{109}$ | $74 \pm _5$ | $459 \pm _{42}$ | $301 \pm _{45}$ | $565 \pm _{54}$ |
| | | Flip | $432 \pm _{55}$ | $315 \pm _{155}$ | $64 \pm _7$ | $409 \pm _{36}$ | $299 \pm _{51}$ | $480 \pm _{49}$ |
| | | Run | $267 \pm _{29}$ | $168 \pm _{60}$ | $26 \pm _2$ | $181 \pm _{21}$ | $113 \pm _{16}$ | $218 \pm _{24}$ |
| | | Stand | $781 \pm _{80}$ | $671 \pm _{113}$ | $168 \pm _{20}$ | $732 \pm _{41}$ | $554 \pm _{66}$ | $871 \pm _{38}$ |
| | | Walk | $606 \pm _{46}$ | $348 \pm _{186}$ | $47 \pm _9$ | $475 \pm _{96}$ | $234 \pm _{57}$ | $654 \pm _{127}$ |
| 1.5x | Cheetah | All tasks | $240 \pm _{286}$ | $13 \pm _{42}$ | $56 \pm _{25}$ | $23 \pm _4$ | $52 \pm _{17}$ | $52 \pm _{55}$ |
| | | Run | $52 \pm _{29}$ | $2 \pm _{41}$ | $17 \pm _{22}$ | $5 \pm _3$ | $18 \pm _{11}$ | $7 \pm _{41}$ |
| | | Run Backward | $24 \pm _{36}$ | $6 \pm _9$ | $11 \pm _9$ | $8 \pm _5$ | $14 \pm _5$ | $15 \pm _{13}$ |
| | | Walk | $715 \pm _{84}$ | $14 \pm _{134}$ | $127 \pm _{92}$ | $29 \pm _{11}$ | $100 \pm _{31}$ | $45 \pm _{204}$ |
| | | Walk Backward | $428 \pm _{290}$ | $18 \pm _{10}$ | $25 \pm _{37}$ | $48 \pm _{18}$ | $74 \pm _{27}$ | $84 \pm _{82}$ |
| | Maze | Multi goal | $250 \pm _{367}$ | $0 \pm _{17}$ | $67 \pm _{41}$ | $39 \pm _{37}$ | $16 \pm _{14}$ | $141 \pm _{50}$ |
| | Quadruped | All tasks | $217 \pm _{79}$ | $177 \pm _{27}$ | $108 \pm _{39}$ | $320 \pm _{26}$ | $264 \pm _{70}$ | $371 \pm _{135}$ |
| | | Jump | $168 \pm _{69}$ | $120 \pm _{78}$ | $74 \pm _{75}$ | $255 \pm _{76}$ | $285 \pm _{96}$ | $297 \pm _{165}$ |
| | | Run | $245 \pm _{139}$ | $119 \pm _{55}$ | $81 \pm _{94}$ | $310 \pm _{101}$ | $200 \pm _{107}$ | $204 \pm _{94}$ |
| | | Stand | $371 \pm _{175}$ | $291 \pm _{46}$ | $155 \pm _{64}$ | $489 \pm _{128}$ | $348 \pm _{140}$ | $546 \pm _{288}$ |
| | | Walk | $190 \pm _{86}$ | $147 \pm _{103}$ | $60 \pm _{37}$ | $252 \pm _{45}$ | $265 \pm _{61}$ | $329 \pm _{103}$ |
| | Walker | All tasks | $364 \pm _{166}$ | $222 \pm _{27}$ | $65 \pm _4$ | $336 \pm _{26}$ | $232 \pm _{50}$ | $514 \pm _{17}$ |
| | | Flip | $273 \pm _{27}$ | $130 \pm _{56}$ | $49 \pm _7$ | $272 \pm _{27}$ | $208 \pm _{44}$ | $384 \pm _{19}$ |
| | | Run | $204 \pm _{40}$ | $82 \pm _{19}$ | $23 \pm _3$ | $136 \pm _{24}$ | $96 \pm _{15}$ | $232 \pm _{24}$ |
| | | Stand | $630 \pm _{88}$ | $461 \pm _{27}$ | $148 \pm _{14}$ | $547 \pm _{17}$ | $419 \pm _{102}$ | $790 \pm _{31}$ |
| | | Walk | $454 \pm _{82}$ | $198 \pm _{63}$ | $38 \pm _{11}$ | $387 \pm _{55}$ | $191 \pm _{53}$ | $641 \pm _{46}$ |
| 2x | Cheetah | All tasks | $312 \pm _{320}$ | $19 \pm _{14}$ | $58 \pm _{35}$ | $23 \pm _7$ | $56 \pm _{19}$ | $24 \pm _{22}$ |
| | | Run | $71 \pm _{55}$ | $6 \pm _{30}$ | $9 \pm _3$ | $5 \pm _1$ | $20 \pm _{14}$ | $8 \pm _{27}$ |
| | | Run Backward | $19 \pm _{37}$ | $6 \pm _{14}$ | $5 \pm _3$ | $9 \pm _5$ | $20 \pm _7$ | $13 \pm _{11}$ |
| | | Walk | $775 \pm _{83}$ | $21 \pm _{31}$ | $147 \pm _{107}$ | $32 \pm _{11}$ | $91 \pm _{30}$ | $43 \pm _{21}$ |
| | | Walk Backward | $552 \pm _{394}$ | $17 \pm _{19}$ | $35 \pm _{64}$ | $48 \pm _{27}$ | $93 \pm _{33}$ | $20 \pm _{79}$ |
| | Maze | Multi goal | $218 \pm _{355}$ | $0 \pm _{11}$ | $65 \pm _{50}$ | $44 \pm _{37}$ | $14 \pm _{12}$ | $131 \pm _{47}$ |
| | Quadruped | All tasks | $215 \pm _{53}$ | $132 \pm _{22}$ | $112 \pm _{27}$ | $270 \pm _{36}$ | $166 \pm _{85}$ | $341 \pm _{116}$ |
| | | Jump | $169 \pm _{140}$ | $62 \pm _{38}$ | $74 \pm _{88}$ | $268 \pm _{79}$ | $170 \pm _{118}$ | $275 \pm _{159}$ |
| | | Run | $221 \pm _0$ | $70 \pm _{66}$ | $86 \pm _{100}$ | $304 \pm _{102}$ | $140 \pm _{96}$ | $179 \pm _{117}$ |
| | | Stand | $315 \pm _{193}$ | $294 \pm _{90}$ | $142 \pm _{30}$ | $353 \pm _{68}$ | $223 \pm _{197}$ | $421 \pm _{210}$ |
| | | Walk | $209 \pm _{122}$ | $70 \pm _{61}$ | $57 \pm _{37}$ | $222 \pm _{78}$ | $111 \pm _{97}$ | $351 \pm _{139}$ |
| | Walker | All tasks | $243 \pm _{118}$ | $157 \pm _{28}$ | $60 \pm _5$ | $186 \pm _{39}$ | $151 \pm _{32}$ | $432 \pm _{24}$ |
| | | Flip | $168 \pm _{40}$ | $105 \pm _{35}$ | $44 \pm _6$ | $149 \pm _{41}$ | $140 \pm _{28}$ | $268 \pm _{32}$ |
| | | Run | $142 \pm _{31}$ | $65 \pm _{13}$ | $22 \pm _4$ | $78 \pm _{20}$ | $69 \pm _{16}$ | $229 \pm _{27}$ |
| | | Stand | $434 \pm _{66}$ | $364 \pm _{53}$ | $143 \pm _{16}$ | $341 \pm _{62}$ | $237 \pm _{51}$ | $656 \pm _{12}$ |
| | | Walk | $319 \pm _{79}$ | $79 \pm _{43}$ | $29 \pm _6$ | $183 \pm _{57}$ | $139 \pm _{49}$ | $567 \pm _{48}$ |

Table 6: **Full results on the ExORL environment generalisation experiments (5 seeds).** For each dataset-domain pair, we report the score at the step for which the all-task IQM is maximised when averaging across 5 seeds $\pm$ the standard deviation. The Baseline FB-GRU represents the scores of an FB-GRU model trained solely on Cheetah-Occluded.

| Environment | Task | FB-stack | **FB-TF (ours)** | **FB-S4d (ours)** | **FB-GRU (ours)** | Baseline FB-GRU |
|---|---|---|---|---|---|---|
| Walker | Walk | $25 \pm 7$ | $34 \pm 8$ | $21 \pm 2$ | $22 \pm 9$ | - |
| | Stand | $126 \pm 28$ | $144 \pm 23$ | $95 \pm 32$ | $82 \pm 18$ | - |
| | Run | $17 \pm 7$ | $26 \pm 5$ | $18 \pm 2$ | $15 \pm 12$ | - |
| | Flip | $23 \pm 8$ | $27 \pm 6$ | $21 \pm 2$ | $23 \pm 10$ | - |
| Quadruped | Walk | $60 \pm 64$ | $99 \pm 39$ | $28 \pm 30$ | $110 \pm 116$ | - |
| | Stand | $150 \pm 79$ | $148 \pm 84$ | $80 \pm 29$ | $257 \pm 128$ | - |
| | Run | $74 \pm 48$ | $71 \pm 26$ | $25 \pm 75$ | $63 \pm 56$ | - |
| | Jump | $71 \pm 35$ | $58 \pm 73$ | $88 \pm 72$ | $190 \pm 92$ | - |
| Cheetah | Walk | $2 \pm 3$ | $24 \pm 12$ | $89 \pm 24$ | $10 \pm 4$ | $38 \pm 216$ |
| | Walk Backward | $12 \pm 6$ | $38 \pm 21$ | $22 \pm 7$ | $10 \pm 12$ | $3 \pm 7$ |
| | Run | - | $3 \pm 2$ | $16 \pm 4$ | $2 \pm 1$ | $8 \pm 46$ |
| | Run Backward | $2 \pm 1$ | $8 \pm 4$ | $4 \pm 1$ | $1 \pm 2$ | $0 \pm 1$ |

