# OpenReview forum: "Foundation Policies with Memory"
_ICLR.cc/2025/Conference — ICLR 2025 Conference Withdrawn Submission_

### Official Review · Reviewer_JGKw · 2024-10-31

**Soundness:** 2
**Presentation:** 3
**Contribution:** 1
**Rating:** 3
**Confidence:** 4

**Summary:**

The paper proposes memory-augmented foundation policies, especially using Transformer, state-space model (S4), and GRU. The authors conduct experiments on POPGYM and ExORL benchmarks comparing the proposed method with FB, which is the backbone model that does not have a memory module, HILP. The results demonstrate that the memory-augmented FB (proposed method) outperforms these baselines.

**Strengths:**

* The paper is generally well-written and easy to understand. Figure 2 provides a clear and direct illustration of the architecture of the proposed method.
* The proposed method, which incorporates a memory module into FB, performs well on the POPGYM and ExORL benchmarks, particularly in the “zero-shot generalization” setting.

**Weaknesses:**

* The biggest concern is the novelty of the proposed method and the experimental design. The method primarily adds a memory module on top of FB [96]. Unlike TMLR and RLC, ICLR considers novelty to be a key criterion, so I cannot give this paper a high rating.

* The authors should clarify why FB [96] was chosen as the backbone model and why FB and HILP are used as baselines. The memory module could be added to other methods, such as HILP, which is the current state-of-the-art. Additionally, FDM [101] performs better than FB, and in some ExORL settings, it even outperforms HILP. To address my concern about the choice of weaker methods as the backbone and baselines, the authors should explore a wider range of methods.

* The authors also need to explain why certain proposed models perform better than others. For example, there is no explanation for why FB-TF performs the worst in POPGym, despite outperforming FB-S4 in ExORL. Moreover, there is a lack of explanation for the misalignment between training and testing performance in Section 4.4. Is this due to the model being over-optimized for the training environment?


* In the problem setting (Section 2), the methods assume that the trajectories collected from RND are highly diverse. However, as the authors acknowledged in Section 5, this is not always the case. They mentioned that limited resources prevented testing in more complex environments. Even several Atari games, such as Montezuma’s Revenge, Gravity, and Jamesbond, could challenge this assumption. For this reason, I am not confident in the proposed method's scalability to more complex environments.


Suggestions:
* Commas are omitted in several places.
For instance, in L193, "In the experiments discussed in Section 4 we" should be "In the experiments discussed in Section 4, we."
* If $x$ in L223 denotes the product operation, please use a different font or simply remove it, as it is confusing. "x" is typically used as a variable, and it is not clearly defined in the paper.
* ICLR has a specific bibliography style, so please ensure the references comply with it.

**Questions:**

* Many state-space models have been published after Diagonalized S4, such as Mamba [Ref 1] and Mamba v2 [Ref 2]. Is there a specific reason for selecting this particular method?
* What is meant by interpolation and extrapolation in Section 4.3?
* Why are there no HILP results for Section 4.4?
* Why are there no results for Maze Top Right, Bottom Left, and Bottom Right in Figure 9?
* Why are there no results for Baseline FB-GRU on Walker and Quadruped in Table 6? Why is Baseline FB-GRU only trained on Cheetah? If that is the case, I believe the authors could add Walker and Quadruped results, as each model is trained on a single environment.
* Is z fixed in each environment once it is sampled, as mentioned in B.1.1?

---

### Official Review · Reviewer_uNK8 · 2024-11-02

**Soundness:** 2
**Presentation:** 3
**Contribution:** 2
**Rating:** 3
**Confidence:** 4

**Summary:**

This work analyzes the behavior of 'foundation policies' for unsupervised RL with recurrent architectures. In particular, it focuses on the forward-backward (FB) algorithm [1]. The authors empirically find that increasing the number of parameters with an additional memory compression mudule (S4/Transformer/GRU) improves performance, with GRU being the most effective choice out of the components explored. Evaluation is carried out on a selected subset of environments from POPGym, and DMC, with some promising generalization results to task settings with varied agent mass and damping coefficient.

[1] Touati, Ahmed, and Yann Ollivier. "Learning one representation to optimize all rewards." Advances in Neural Information Processing Systems 34 (2021): 13-23.
[2] Morad, Steven, et al. "Popgym: Benchmarking partially observable reinforcement learning." arXiv preprint arXiv:2303.01859 (2023).
[3 Tassa, Yuval, et al. "Deepmind control suite." arXiv preprint arXiv:1801.00690 (2018).]

**Strengths:**

1. Unsupervised RL and environment generalization are ambitious open problems that I believe to be very relevant to improving RL toward being more applicable for real-world use cases.

2. The authors provide their implementation and highlight the specific changes they made from the implementation in the prior FB papers, which appears to make their contributions highly reproducible.

3. Overall, I found the paper well-structured and easy to read.

**Weaknesses:**

Major:

1. The main setting used to evaluate 'test-time' generalization simply corresponds to varying slightly the agent's mass and damping coefficient. Some results are also shown on the occluded-DMC environments (for which I could not find almost any details/precise specification in either the paper or the appendix, and had to refer to the code), but the performance of all methods does not appear to be statistically significantly different than random. I believe these settings and results to be quite underwhelming, especially in contrast to the repeated claims of improved generalization "on entirely new environments not encountered during training." (e.g., line 22). In my opinion, I would suggest toning down these claims or, again, expanding experimentation. For instance, I believe including test environments where agents have similar configurations but different morphologies, or trying to improve the performance on the occluded tasks by training on more environments is crucial to improve this work.

2. Overall, I also found the environments examined to be relatively easy and lack challenge. In particular, the authors only evaluate from a few proprioception-based tasks comprising relatively toyish environments from POPGym, and a few other relatively simple environments from DMC. I believe this to be insufficient for a paper that does not make any theoretical or algorithmic contribution, and focuses on empirically comparing the interaction of existing methodologies. To provide more informative and meaningful results, I would suggest also analyzing harder environments (e.g., Jaco, Humanoid) or even working from pixel observations.

3. Given the paper's focus is comparing different architectures, each with a distinct set of hyperparameters, I was also quite disappointed with 1) Lack of any thorough hyperparameter study for the different methods. 2) Only employing 5 random seeds.

Minor:

1. In line 239 the statement "FB is the most performant FP utilizing successor measures, and HILP is the most performant FP utilizing successor features" and in line 263 the statement "GRU [...] is the most performant RNN on the POPGym benchmark" lack either a reference to results and/or other literature.

2. I am not sure about the actual meaningfulness and analysis of the interpolation/extrapolation-to-train ratios in Section 4.3 (Figure 4). FB-TF seems to obtain higher interpolation than training performance which I would believe is likely indicative that the low number of seeds used, makes stochasticity play a major part in performance. I would encourage the authors to move these parts to the Appendix and address or explicitly acknowledge these areas of improvement in future revisions.

**Questions:**

I would appreciate it if the authors could address the criticism raised above. In addition, could the authors clarify if the hyper-parameters for the different recurrent methods were swept in any way? (if this is the case, I would advice including these results in the paper or Appendix).

---

### Official Review · Reviewer_hod9 · 2024-11-03

**Soundness:** 4
**Presentation:** 2
**Contribution:** 3
**Rating:** 3
**Confidence:** 2

**Summary:**

This paper introduces an enhancement to Foundation Policies (FPs) by integrating memory models to enable better performance in unknown environments. While traditional FPs can generalize across tasks with similar dynamics after pre-training, they struggle with new tasks that have different dynamics. By incorporating memory architectures such as attention mechanisms, state-space models, and recurrent neural networks (RNNs) like GRUs, the authors allow FPs to adapt based on inferred dynamics. Evaluation on memory (POPGym) and unsupervised RL (ExORL) benchmarks reveals that GRUs notably improve FP adaptability, nearly matching supervised baselines and outperforming memory-free FPs. This approach advances the creation of adaptable, generalist agents capable of zero-shot learning in unseen environments.

**Strengths:**

- Well-implemented and straightforward idea to have memory within FPs.

**Weaknesses:**

- Limited exploration of memory models: it is not clear to me why GRU would be better memory models than state-space models (S4) and transformers (TF). Hyper-parameter tuning for each architecture is not very clear and GRU might have had more hyper-parameter optimization than the other methods. Could you provide a more detailed description of how hyperparameters were selected for each architecture, or include ablation studies showing the impact of key hyperparameters?
- Unclear whether the proposed solution allows better generalization to completely unseen environments with very different dynamics and rewards. Would it be possible to include a discussion of the limitations of your current evaluation in terms of environment diversity?

**Questions:**

1. How would you qualify the level of distribution shift tested with POPGym and ExORL? It isn't clear how much OOD generalization is actually tested here.
2. You're saying that with a higher computational budget, you would set L=H (the horizon of an episode). For architectures like the transformers where probabilistic inference grows quadratically with the context size, do you think it would still be possible to increase L to very high values?

---

### Official Review · Reviewer_iC3y · 2024-11-04

**Soundness:** 2
**Presentation:** 2
**Contribution:** 2
**Rating:** 3
**Confidence:** 4

**Summary:**

The paper mainly augments the foundation policies, especially the Hilbert foundation policies, with memory-based architectures like RNN and attention. Intuitively, the proposed method replaces the state $s$ with the memory $h$ in the feature function $\psi$ and the subsequent policy $\pi$. Experiments on three benchmarks demonstrate the superiority of the memory-based architecture over vanilla baselines.

**Strengths:**

- The topic of foundation policies meets the urgent trend of scaling RL algorithms towards generalist models.
- Experiments are conducted on three representative benchmarks.

**Weaknesses:**

- The way of simply using memory-based architecture to improve the vanilla algorithm is trivial and widely investigated. The memory, e.g., h in the paper, is a temporal abstraction of the history, and inevitably contains more useful information than the single-step state s itself.
- Experiments are not well prepared. The proposed memory-based architecture is just compared to the vanilla foundation policy baseline. As mentioned above, this superiority is taken for granted, and hence cannot justify the key contribution of the proposed method and its advantages over other strong baselines.
- The assumption of the proposed method can in turn limit its capability towards larger-scale deployments. It assumes the reward function is linear w.r.t. the state feature \varphi(s). The linearity will limit the expressivity of the function approximation, especially in complex problems or with large-scale architectures. Since the goal of foundation policies is to build generalists, probably with the scaling law, this assumption could be a tough limit for the proposed method.

**Questions:**

- In the first paragraph, “FPs are not equipped to deal with a change in dynamics between pre-training and deployment”, is there any evidence to support this claim? It is kind of counter-intuitive. The goal of any foundation models is to develop generalists that can be transferred to unseen downstream tasks, with little to no fine-tuning.
- In the second paragraph, it claims that “in-context RL cannot reliably generalize to new tasks with different functions”, which is questionable. The goal of in-context learning is to pre-train a foundation model that can be directly deployed to downstream or unseen tasks with only some prompt examples. Like in the assumption of meta-RL, the meta-policy is tested in those tasks that are excluded from training ones, while the test tasks can be sampled from the same distribution as the training ones.
- In eq. 2, computing policy \pi involves an optimization loop of finding the action that maximizes $\psi(s,a,z)^Tz$. For discrete action spaces, this computation is straightforward. What about for continuous action spaces? Is an inner optimization loop needed for this?
- In the FP backbone, the $z_{test}$ is computed using an optimization loop w.r.t. variable z. What is the computation cost if z is a continuous variable?

---

### Note · Authors · 2024-11-25

**Comment:**

Thanks to the reviewers for their feedback and comments. Clearly we have to rework the paper, and we intend to use more time than the discussion period allows to make the required changes. As a result we are issuing a withdrawal. Thanks again for your feedback.

**Withdrawal Confirmation:**

I have read and agree with the venue's withdrawal policy on behalf of myself and my co-authors.